

# A redescription of *Palaeogekko risgoviensis* (Squamata, Gekkota) from the Middle Miocene of Germany, with new data on its morphology

Andrea Villa

Institut Català de Paleontologia Miquel Crusafont, Cerdanyola del Vallès, Barcelona, Spain

## ABSTRACT

After its original description, the Middle Miocene gekkotan *Palaeogekko risgoviensis* remained an enigma for palaeontologists due to a rather poor knowledge of its osteology and relationships. Coming from a single locality in southern Germany, this gecko lived in central Europe during a period when a single gekkotan lineage (*i.e.*, euleptine sphaerodactylids) is confidently reported to have inhabited the continent. However, it is unclear whether *P. risgoviensis* may represent a member of this same lineage or a second clade of Gekkota. In order to shed light on this issue, the type material of *P. risgoviensis* is here redescribed, refigured and extensively compared with extinct and extant geckos from Europe. A phylogenetic analysis is also conducted in order to investigate its relationships. The new observations confirm the validity of the German species as a distinct taxon, and exclude the previously-suggested chimeric status of the type material of this gecko (with the exception of a single dentary included in the type series, which clearly belong to a different lizard). Phylogenetic relationships of *Palaeogekko* are still unclear, though, with different positions within the gekkotan tree recovered for the taxon. Nevertheless, it is confidently supported as a non-eublepharid gekkonoid, in agreement with hypothesys presented by other scholars.

## INTRODUCTION

The beginning of the Miocene Epoch is considered to mark the moment when the current European squamate fauna started assembling (*Rage, 2013*; *Villa & Delfino, 2019b*; *Georgalis & Scheyer, 2021*), with fossils related to extant species or species complexes even reported already from the Early Miocene (for lizards, see *e.g.*, *Čerňanský, 2010*; *Venczel & Hír, 2013*). Changes in the environment around the Oligocene/Miocene transition were the prelude to a subsequent faunal turnover facilitated by the establishment of a stable connection between Eurasia and Africa and the return to a warm and humid climate during the Miocene Climatic Optimum, both events occurring in the Early Miocene (*Böhme, 2003*; *Rage, 2013*; *Georgalis, Villa & Delfino, 2016*). The turnover saw the appearance in Europe of several taxa that characterized the European reptilian fauna in the Neogene (*e.g.*, chameleons: *Georgalis, Villa & Delfino, 2016*; glass lizards of the genus

Corresponding author
Andrea Villa, andrea.villa@icp.cat

*Pseudopus Merrem, 1820*: *Čerňanský, Rage & Klembara, 2015*; *Vasilyan et al., 2019*; *Villa, Gobbi & Delfino, 2022*; monitor lizards: *Ivanov et al., 2018*; *Villa et al., 2018a*; cobras: *Szyndlar & Schleich, 1993*; vipers: *Szyndlar & Rage, 1999*, *2002*), and in most cases survived in the continent through the Quaternary and up to today.

A possible exception to the Miocene roots pattern is modern European geckos, given that fossils of the most widespread extant taxa are not known from fossiliferous deposits older than the Upper Pliocene (*Villa & Delfino, 2019b*). Only the sphaerodactylid *Euleptes Fitzinger, 1843* has its first appearance in the Early Miocene (*Müller, 2001*; *Müller & Mödden, 2001*; *Čerňanský & Bauer, 2010*). Other sphaerodactylids related to the extant western Mediterranean endemic *Euleptes europaea* (*Gené, 1839*) were present in Europe at least from the Eocene (*Villa, Wings & Rabi, 2022*), including several genera and species that represented the dominant clade of gekkotans in Europe for most of the Cenozoic. This is in contrast to the modern situations that sees phyllodactylids and gekkonids as the most widespread geckos north of the Mediterranean Sea. Members of this sphaerodactylid lineage in the Miocene included at least two other *Euleptes* species and two species referred to the genus *Gerandogekko Hoffstetter, 1946*.

The phylogenetic relationships of a fifth Miocene species, *Palaeogekko risgoviensis Schleich, 1987*, are unclear. It was described as a gekkonine gekkonid by *Schleich (1987)*. At that time, Gekkonidae was used in a wide sense, including all gekkotans exclusive of pygopodids, but *Schleich (1987)* explicitly mentioned affinities with *Tarentola Gray, 1825* and *Cyrtodactylus Gray, 1827* (in fact, *Mediodactylus Szczerbak & Golubev, 1977*, given that he was using *Cyrtodactylus kotschyi*, now part of the other genus, for comparisons). Thus, *P. risgoviensis* would be related to either phyllodactylids or gekkonids (in a modern sense) according to his observations. Later on, *Daza, Bauer & Snively (2014)* commented on the species, stating that the available information only allowed to exclude belonging to pygopodoids and eublepharids, in agreement with previous conclusions, but preventing further discrimination between other clades within gekkonoids. More recently, a phylogenetic analysis by *Villa, Wings & Rabi (2022)* recovered *P. risgoviensis* as an unstable taxon, but resulting topologies possibly placed it either as a crown sphaerodactylid or as a stem pygopodoid. Considering all the possibilities suggested by the different authors, *P. risgoviensis* may either represent another species related to the dominant European sphaerodactylid lineage or the first evidence of a second gekkotan clade in Europe during the Miocene.

Until now, clarification of the taxonomy and phylogenetic affinities of *P. risgoviensis* was hampered by a somehow scarce knowledge of its osteology, which has never been revised after *Schleich's (1987)* original article. The original description was rather synthetic and mostly focused on presenting measurements and meristic features, and only part of the material was properly figured. Thus, several aspects of the precise morphology of this gecko remained difficult to assess. Moreover, our understanding of gekkotans osteology significantly increased since the late 80s, in particular as far as the extant and extinct European taxa are concerned (for these, see *e.g.*, *Daza, Bauer & Snively, 2014*; *Villa et al., 2018b*). In light of all of this, a redescription of the type material referred to *P. risgoviensis* under modern knowledge of gekkotan osteology is here presented. Additionally, detailed

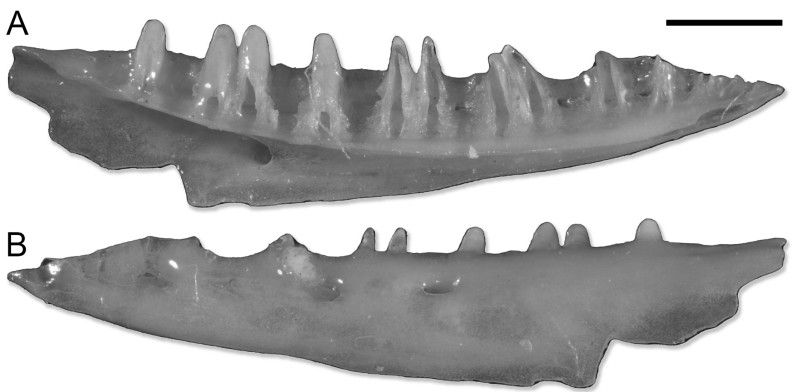

**Figure 1  SNSB-BSPG 1970 XVIII 7262, left dentary of Lacertoidea vel Scincoidea indet.** (A) Medial view. (B) Lateral view. Scale bar equals 1 mm. Pictures of the specimen were taken by Victor Beccari.

comparisons particularly focused on other European gekkotans and an attempt at better understanding the possible phylogenetic position of this Middle Miocene gecko are also provided. This will make available new useful data also for future studies further investigating the diversity and evolution of geckos in Europe and nearby continents.

## MATERIALS AND METHODS

The type material of *P. risgoviensis* is housed in the Bayerische Staatssammlung für Paläontologie und Geologie (SNSB-BSPG), in Munich (Germany). *Schleich (1987)* originally referred 118 specimens to his new species, including the holotype lower jaw, 90 dentaries, 23 maxillae, and four premaxillae. However, one of the dentaries, SNSB-BSPG 1970 XVIII 7262 (Fig. 1), has an open Meckelian fossa and cannot pertain to a gecko. Furthermore, it was not possible to clearly locate two other dentaries, SNSB-BSPG 1970 XVIII 7249 and 7270, in the available material. A single unnumbered specimen stored with the rest of the type series most likely represents the former, even though measurements and tooth count do not exactly agree with those reported by *Schleich (1987*: tab. 1). Revised measurements and counts of teeth and foramina reported in the descriptions are based on the best-preserved specimens. Detailed data for all specimens are provided in the Data S1.

The anatomical terminology used in this article follows *Villa et al. (2018b)* and *Villa & Delfino (2019a)*. Measurements were taken with a digital caliper. Selected specimens were photographed with a Leica M165 FC microscope equipped with a DFC450 camera and the Leica Application Suite (LAS) 4.5. *Mediodactylus kotschyi* (*Steindachner, 1870*) is here used in a wide sense, without separating it into the several species now recognised (see *Speybroeck et al., 2020*). This is not in opposition to the split, but simply to ease comparisons with the most-widespread European extant geckos as defined by *Villa et al. (2018b)*. Comparisons are based on the same specimens of extant gekkotans used by *Villa et al. (2018b)*, as well as on available literature and personal observations (for *Gerandogekko arambourgi Hoffstetter, 1946* and *Geiseleptes delfinoi Villa, Wings & Rabi, 2022*) for extinct taxa.

For the phylogenetic analysis, *Villa, Wings & Rabi's (2022)* matrix was used, which is available in MorphoBank (*O'Leary & Kaufman, 2012*) at the following link: http://morphobank.org/permalink/?P4069. *Palaeogekko risgoviensis* was rescored in Mesquite 3.70 (*Maddison & Maddison, 2021*) after the new morphological information provided by the updated description herein presented (see the Data S2 for the revised scorings). An unweighted maximum parsimony analysis was run in TNT 1.5 (*Goloboff, Farris & Nixon, 2008*; *Goloboff & Catalano, 2016*). As in the second iteration of the analysis by *Villa, Wings & Rabi (2022)*, G. arambourgi was removed and the same six constraints were applied (all fossils retained as floating taxa). These constraints follow the topology recovered from molecular data by *Gamble et al. (2015)*. The matrix was analysed using the New Technology search, with all options selected and the consensus stabilized five times with a factor of 75. The "Collapse trees after search" option was selected to avoid zero-length branches in the individual resulting trees. A second round of tree bisection and reconnection was run after the first New Technology search.

## Systematic palaeontology

Order Squamata *Oppel, 1811*
Infraorder Gekkota *Camp, 1923*
Superfamily Gekkonoidea *Gray, 1825*
Family incertae sedis
Genus *Palaeogekko Schleich, 1987*
Species *Palaeogekko risgoviensis Schleich, 1987*

**Holotype:** SNSB-BSPG 1970 XVIII 7300, a complete right lower jaw.

**Paratypes:** four premaxillae (SNSB-BSPG 1970 XVIII 7363/7366); 23 maxillae (SNSB-BSPG 1970 XVIII 7340/7362); 90 dentaries (SNSB-BSPG 1970 XVIII 7249/7299, 7301/7339).

**Type locality and age:** Steinberg, Nördlinger Ries, southern Germany; Middle Miocene, MN 6 (*Heizmann & Fahlbusch, 1983*; *Prieto & Rummel, 2016*).

**Emended diagnosis:** *Palaeogekko risgoviensis* is diagnosed by the following combination of characters: (1) narrow ascending nasal process of the premaxilla, with a shallow expansion at midheight; (2) shallow notch separating the palatal processes of the premaxilla; (3) absence of a groove following the last ventrolateral foramen on the maxilla; (4) short and pointed posterior process of the maxilla; (5) presence of a sigmoid and more-or-less vertically-directed carina maxillaris on the maxilla; (6) anterior mylohyoid foramen present as a notch on the splenial; and (7) posterior surangular foramen not shifted dorsally.

**Remarks**

The dentary SNSB-BSPG 1970 XVIII 7262 (Fig. 1) does not actually pertain to a gecko. The combination of a completely open Meckelian fossa, a narrow subdental shelf, and the

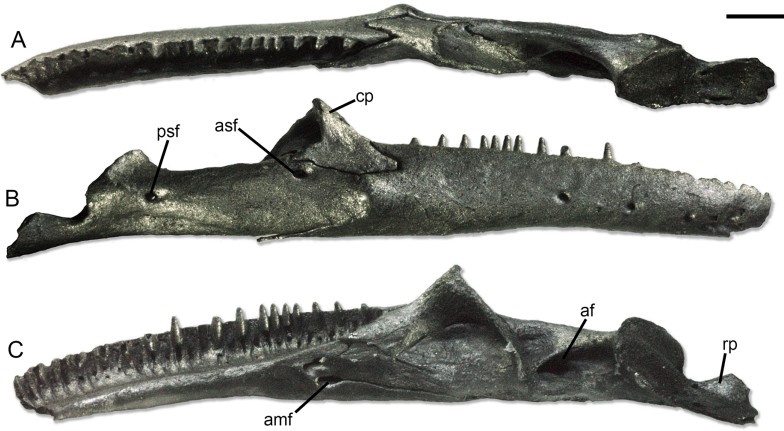

**Figure 2 Holotype right lower jaw (SNSB-BSPG 1970 XVIII 7300) of *Palaeogekko risgoviensis*.** (A) Dorsal view. (B) Lateral view. (C) Medial view. Scale bar equals 1 mm. Abbreviations: af, adductor fossa; amf, anterior mylohyoid foramen; asf, anterior surangular foramen; cp, coronoid process; psf, posterior surangular foramen; rp, retroarticular process.

pleurodont dentition suggests it belongs to either a lacertoid or scincoid lizard, but the preservation prevents a more precise referral. This specimen is here listed within the paratypes of *P. risgoviensis*, in agreement with *Schleich (1987)*, but it is not considered in the descriptions and the rest of the article.

**Description of the holotype**

The holotype SNSB-BSPG 1970 XVIII 7300 (Fig. 2) is an almost complete right lower jaw, missing only the symphyseal region of the dentary. The overall shape of the jaw is rather slender and straight. On the lateral side, a knob-like swelling is visible at the level of the contact between the compound bone and the dentary. The swelling partially obscures the suture between the two skeletal elements. It may have a pathological origin, as already noted by *Schleich (1987)*. The preserved length of the jaw is 13 mm; the length of dentary tooth row is at least 6.38 mm (not considering the missing portion). The dentary carries 30 tooth positions in the preserved part.

**Dentary.** The dentary is narrow and elongated, with a straight ventral margin. In medial view (Fig. 2C), the Meckelian fossa is almost completely closed in a narrow tubular structure, which opens posteriorly in a U-shaped notch. The extension of this notch related to the alveolar shelf is not clearly measurable, but it was likely one fifth or smaller. Dorsally, a low subdental ridge is present, as well as a wide and deep sulcus dentalis. The dentary bears a long, narrow and pointed inferior posterior process and a short and forked superior posterior process. In the superior process, the ventral projection seems slightly longer than the dorsal one. The lateral surface is smooth (Fig. 2B), with at least four mental foramina; most probably, the number of foramina was higher in origin, considering the broken anterior end of the bone.

**Splenial.** The splenial (Fig. 2C) is preserved, even though damaged. It is a small and slender splint of bone, with pointed anterior and posterior ends. The posterior end of the bone does not bend ventrally to the jaw. Anteriorly, the anterior mylohyoid foramen is clearly represented by a long and shallow notch along the ventral margin of the bone. The smaller anterior inferior foramen is also visible as a notch in the specimen as it is preserved, but this could be the result of a clear breakage of the anterodorsal part of the splenial.

**Coronoid.** The coronoid is strongly concave in medial direction (Fig. 2A). The anterior part of the bone is composed by a long and wide anteromedial process and a shorter (roughly half the length of the former) labial process. Both processes are pointed. The coronoid process is straight and dorsally pointed. The posteromedial process is long and slender, with a pointed end. An osseous lamina connects the coronoid process and the posteromedial process on the medial side of the bone.

**Compound bone.** The compound bone is completely fused. On the medial side (Fig. 2C), the adductor fossa is rather narrow and anteroposteriorly elongated. Ventral to the anterior end of the fossa, a very low osseous expansion in visible in dorsal view (Fig. 2A), even though it is not clear whether this could be pathological in nature or not. The articular condyle is subcircular in posterodorsal view, with a longitudinal swelling in the middle. The swelling divides the articular surface into two areas, the lateral one slightly larger than the medial one. The base of the retroarticular process is moderately slender, but the process expands posteriorly. The posterior margin of the retroarticular process is broken off, preventing a clear recognition of its complete shape. Nevertheless, the expansion appears rather strong. On the dorsomedial side of the process, the foramen for the chorda tympani is wide and elongated. A second, smaller foramen is also visible by the anterodorsal corner of the medial surface of the retroarticular process. A distinct lateral crest is not visible on the preserved lateral surface of the process. The lateral surface of the compound bone is generally smooth (Fig. 2B). Ventral to the contact with the coronoid, there is a wide anterior surangular foramen, opening in lateral direction. This foramen is completely within the compound bone, with no contribution of neither the coronoid or the dentary to its borders. There is no groove associated to it. The posterior surangular foramen is not shifted dorsally, opening more or less at midheight of the bone. A low ridge running from the posterior surangular foramen to the articular condyle is present. Ventral to the same foramen, a shallow but distinct sunken area is visible, being narrow and anteroposteriorly elongated. There is no posterior mylohyoid foramen.

**Dentition.** Teeth (Figs. 2 and 3A) are pleurodont and homodont. They are closely spaced, narrow and subcylindrical. They narrow distinctly towards the crown, ending with a pointed tip. They are exposed laterally for about one third of their height. The crown is bicuspid, with a labial and a lingual cusps. There is no striation neither labially nor lingually.
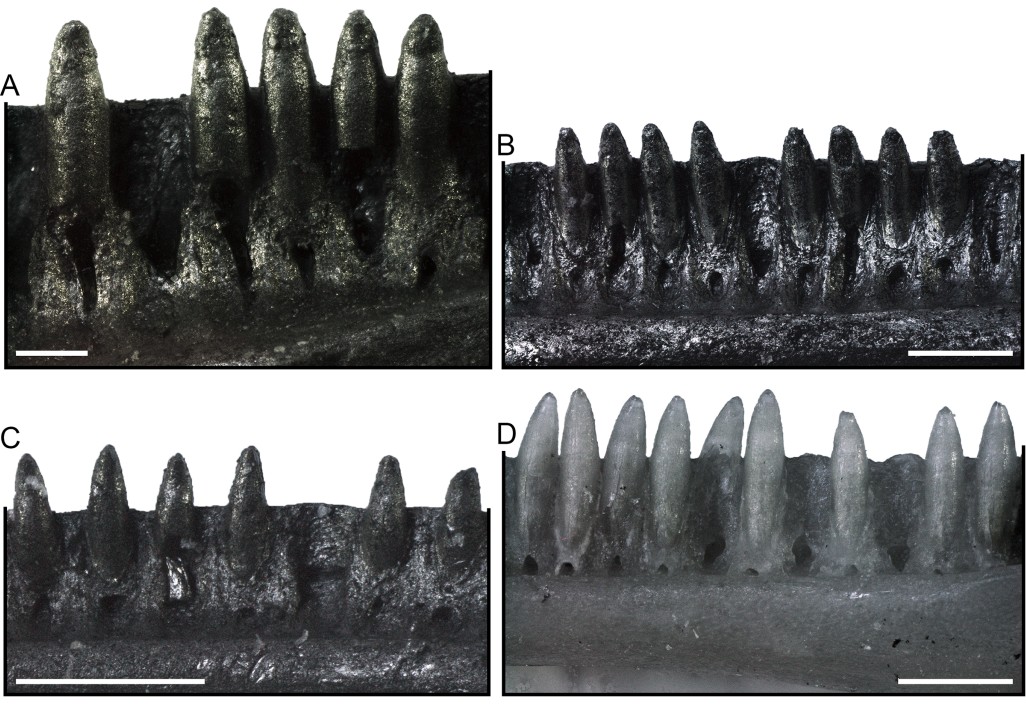

**Figure 3 Dentition of *Palaeogekko risgoviensis*.** (A) SNSB-BSPG 1970 XVIII 7300 (Holotype). (B) SNSB-BSPG 1970 XVIII 7252. (C) SNSB-BSPG 1970 XVIII 7299. (D) SNSB-BSPG 1970 XVIII 7301. All in lingual view. Scale bars equal 0.2 mm (A) and 0.5 mm (B–D).

### Description of the paratypes

All paratypes are moderately small and slenderly built.

**Premaxillae (Fig. 4).** The unpaired premaxillae bear well-developed palatal processes, which are separated posteromedially by a shallow and wide notch. Thus, the processes form a long subtrapezoidal lamina extending posteriorly from the alveolar portion of the bone. The width of the alveolar portion varies between 2.08 to 2.21 mm. The ascending nasal process is moderately long and narrow. It has a more or less constant width (slightly narrowing by the tip only in SNSB-BSPG 1970 XVIII 7366; Figs. 4G and 4H), showing only a variable but always poorly-developed expansion roughly at midheight. The end of the process appears rounded, not pointed, but it is not clear if this could be just an artifact due to breakage. The anterior surface is smooth, whereas the posterior one is characterized by the presence of the septonasal crest. This crest is well distinct and blunt in the ventral half of the process. In the dorsal half, it varies from absent, to very poorly visible, to distinct and sharp. In the same portion of the process, the long and narrow articular surfaces for the anteromedial processes of the nasals are visible by the sides of the crest. The foramina for the longitudinal canals are small. By each side of the ascending nasal process, these foramina are associated to an accessory foramen, which is located dorsal to the former. Only on the right side of SNSB-BSPG 1970 XVIII 7366, this accessory foramen is absent.

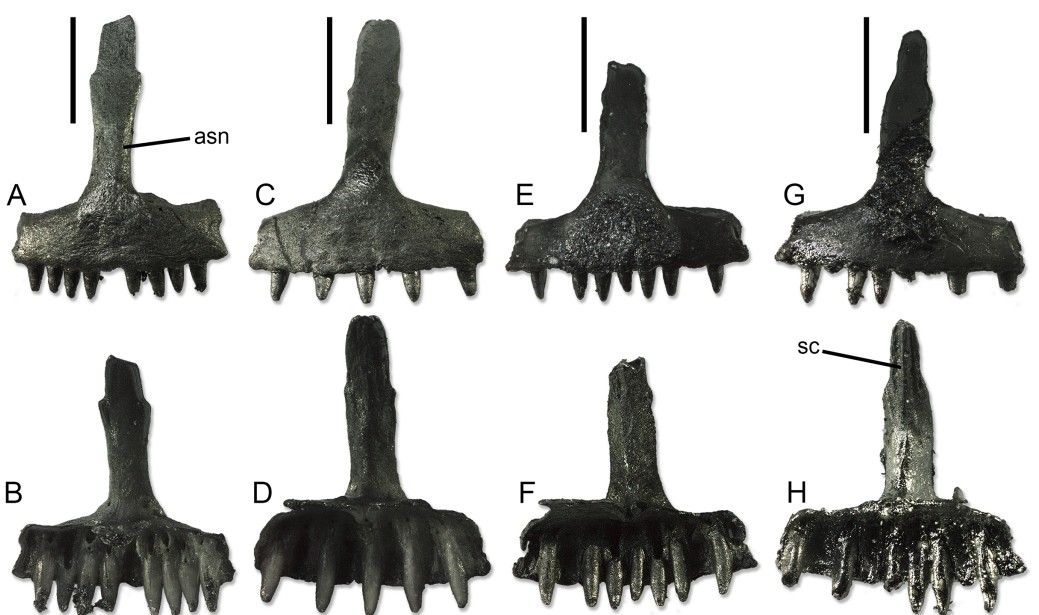

**Figure 4 Paratype premaxillae of *Palaeogekko risgoviensis*.** (A and B) SNSB-BSPG 1970 XVIII 7363. (C and D) SNSB-BSPG 1970 XVIII 7364. (E and F) SNSB-BSPG 1970 XVIII 7365. (G and H) SNSB-BSPG 1970 XVIII 7366. (A, C, E and G) Anterior views. (B, D, F and H) Posterior views. Scale bars equal 1 mm. Abbreviations: asn, ascending nasal process; sc, septonasal crest.

**Maxillae (Fig. 5).** In medial view, the tooth row runs along the whole length of the maxilla. The length of the row in the best-preserved specimens ranges from 5.88 to 6.96 mm. The row of a possibly juvenile specimen, SNSB-BSPG 1970 XVIII 7361 (Fig. 5S), is 4.56 mm long. Anteriorly, the maxilla displays a well-developed anterior premaxillary process. In dorsal view, the latter displays a shallow, U-shaped anterior concavity (Figs. 5G and 5P). The anterolateral process varies from almost absent to short, whereas the anteromedial one is more developed. When developed, the anterolateral process is anteriorly truncated. The anteromedial one, on the other hand, is more pointed, at least based on the only specimen in which it is well preserved, SNSB-BSPG 1970 XVIII 7362; specimens where the process appears truncated are indeed present, but a close inspection suggests that the process is broken in these cases. The moderately wide vomeronasal foramen is located in the middle of the dorsal surface of the anterior premaxillary process. The facial process is roughly half as long as the entire maxilla or slightly less than that. It is laminar and has smooth medial and lateral surfaces. On the medial side, a very low and more-or-less sigmoid carina maxillaris (medial ridge *sensu Villa et al., 2018b*) is present close to the anterodorsal corner (Figs. 5A, 5E and 5Q). The main course of the carina is vertically oriented, not inclined. The anterior margin of the process is straight, slightly inclined towards the anterior. This inclination originates a short point extending from the top of the margin. The anterodorsal margin is gently oblique. The posterodorsal part of the process is missing or strongly damaged in all specimens, but the preserved portion of the posterior margin in at least some of them suggests that this was straight and steeply inclined (note that this morphology was figured in the drawings by *Schleich (1987)*: fig. 6).

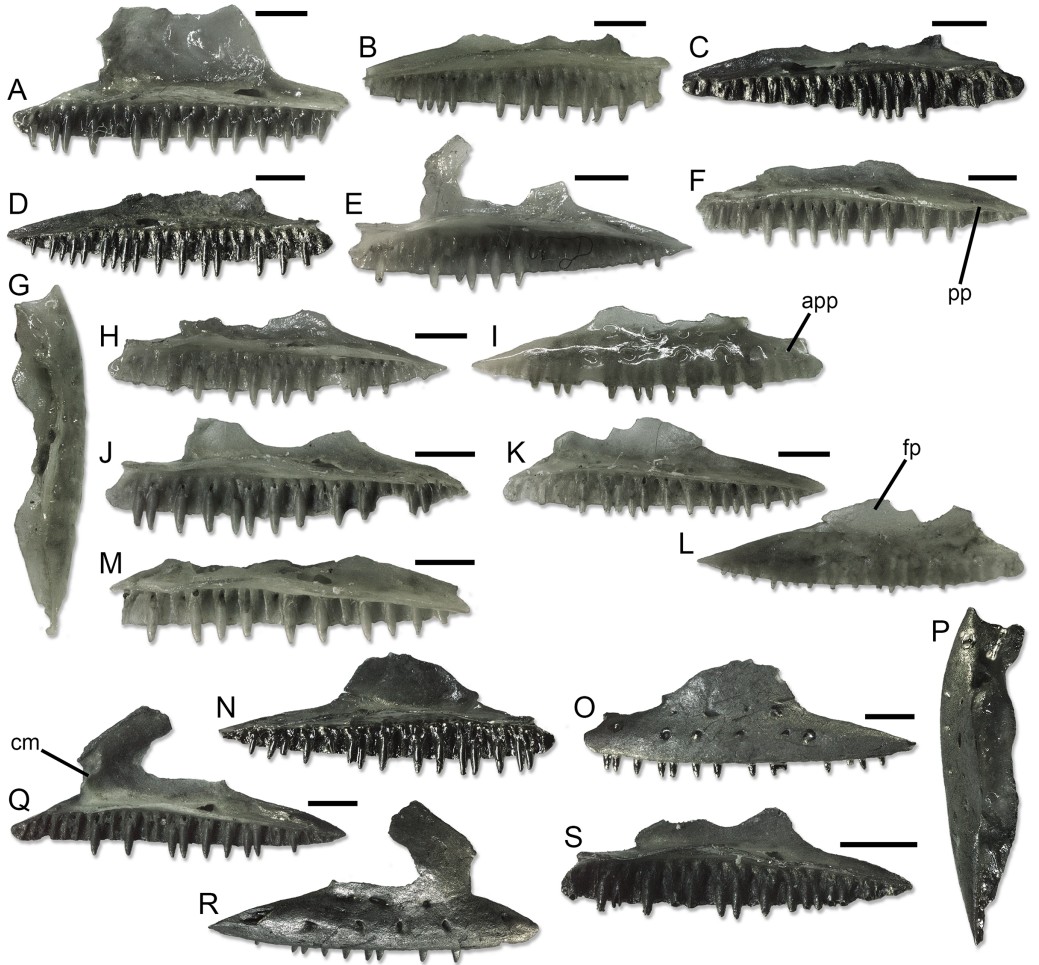

**Figure 5 Paratype maxillae of *Palaeogekko risgoviensis*.** (A) SNSB-BSPG 1970 XVIII 7344. (B) SNSB-BSPG 1970 XVIII 7348. (C) SNSB-BSPG 1970 XVIII 7350. (D) SNSB-BSPG 1970 XVIII 7351. (E) SNSB-BSPG 1970 XVIII 7352. (F) SNSB-BSPG 1970 XVIII 7353. (G–I) SNSB-BSPG 1970 XVIII 7354. (J) SNSB-BSPG 1970 XVIII 7355. (K and L) SNSB-BSPG 1970 XVIII 7356. (M) SNSB-BSPG 1970 XVIII 7357. (N–P) SNSB-BSPG 1970 XVIII 7359. (Q and R) SNSB-BSPG 1970 XVIII 7360. (S) SNSB-BSPG 1970 XVIII 7361. (A–F, H, J, K, M, N, Q and S) Medial views. (G and P) Dorsal views. (I, L, O and R) Lateral views. Scale bars equal 1 mm. Abbreviations: app, anterior premaxillary process; cm, carina maxillaris; fp, facial process; pp, posterior process.

The superior dental foramen is located dorsally on the palatal shelf, by the end of the facial process. It is wide and opens posteriorly, continuing on a shallow and wide groove on the posterior process. There is no lacrimal groove. The posterior process is pointed and rather short, never exceeding the facial process in length. There is no longitudinal groove following the last ventrolateral foramen laterally. On the lateral surface of the maxilla, the ventrolateral foramina range from five to eight in number. A second row made up by two to seven foramina is also present dorsal to the ventrolateral ones, by the base of the facial process.

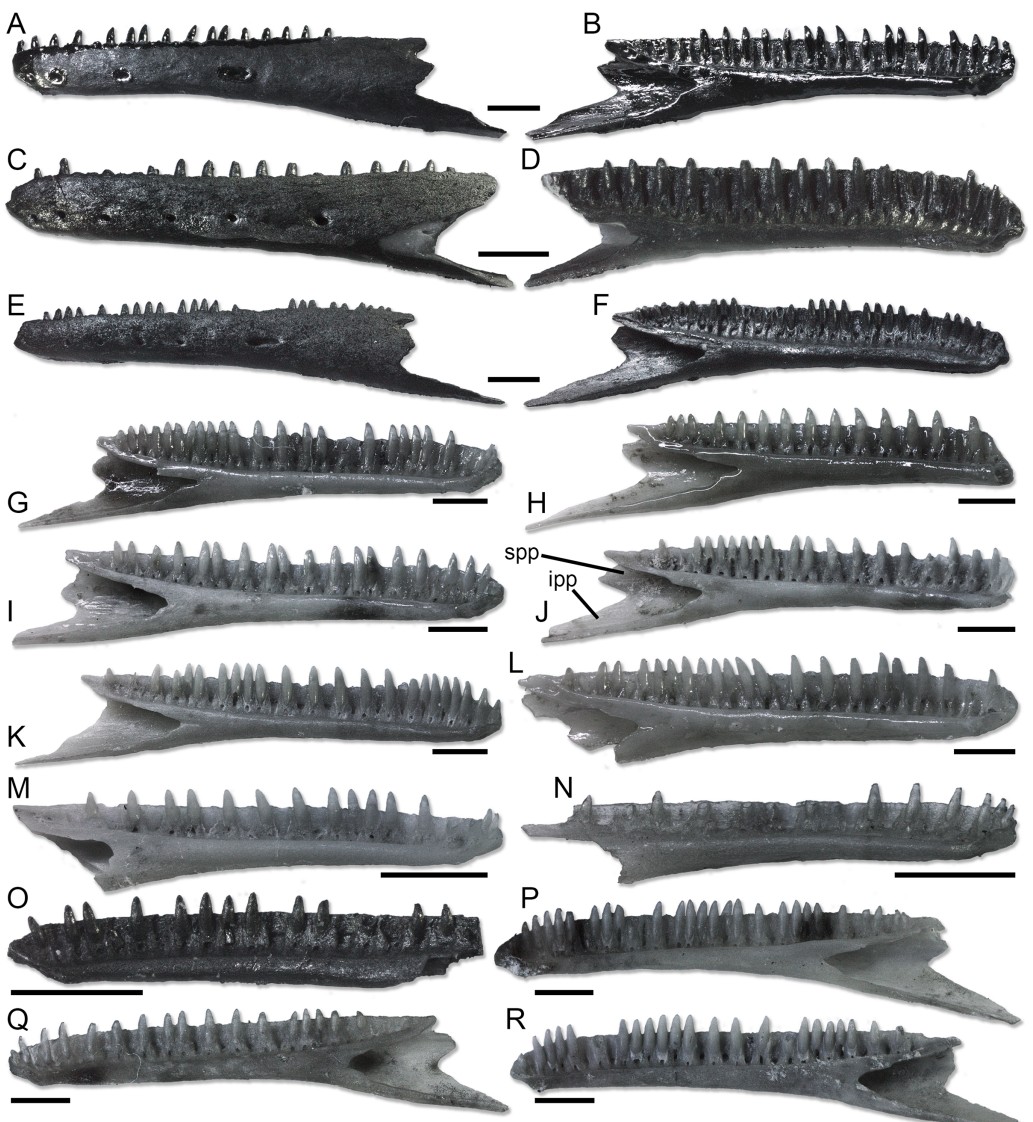

**Figure 6 Paratype dentaries of *Palaeogekko risgoviensis*.** (A and B) SNSB-BSPG 1970 XVIII 7250. (C and D) SNSB-BSPG 1970 XVIII 7251. (E and F) SNSB-BSPG 1970 XVIII 7252. (G) SNSB-BSPG 1970 XVIII 7253. (H) SNSB-BSPG 1970 XVIII 7254. (I) SNSB-BSPG 1970 XVIII 7257. (J) SNSB-BSPG 1970 XVIII 7259. (K) SNSB-BSPG 1970 XVIII 7263. (L) SNSB-BSPG 1970 XVIII 7274. (M) SNSB-BSPG 1970 XVIII 7290. (N) SNSB-BSPG 1970 XVIII 7291. (O) SNSB-BSPG 1970 XVIII 7299. (P) SNSB-BSPG 1970 XVIII 7301. (Q) SNSB-BSPG 1970 XVIII 7302. (R) SNSB-BSPG 1970 XVIII 7307. (A, C and E) Lateral views. (B, D and F–R) Medial views. Scale bars equal 1 mm. Abbreviations: ipp, inferior posterior process; spp, superior posterior process.

**Dentaries (Fig. 6).** Paratype dentaries generally resemble the homologous bone of the holotype in morphology. They are slenderly built, with a closed Meckelian fossa. The tubular structure closing the fossa narrows anteriorly and opens posteriorly with a V-shaped (rarely U-shaped) notch that is about as long as one fifth of the tooth row. Only in two specimens, SNSB-BSPG 1970 XVIII 7308 and SNSB-BSPG 1970 XVIII 7310, this notch is shorter, about one seventh and one sixth of the tooth row length respectively. Anteriorly, the mandibular symphysis is narrow and slightly inclined in dorsal direction.

Ventral to the symphysis, the Meckelian fossa opens on the ventral surface of the dentary in a short longitudinal groove. The dentaries display a distinct subdental ridge (less defined in some cases: *e.g.*, SNSB-BSPG 1970 XVIII 7252), marking a deep and wide sulcus dentalis. The inferior posterior process is long and pointed. The superior posterior process is short and forked, being made up by two triangular projections separated by a notch. When preserved, the ventral projection is longer than the dorsal one. The lateral surface of the dentaries is smooth, with four to seven mental foramina. The ventral margin is straight in medial and lateral views. The length of the tooth row in the best-preserved specimens goes from 3.92 to 7.81 mm.

**Splenial.** SNSB-BSPG 1970 XVIII 7281 also preserves a fragment of splenial, which is partially fused with the dentary. The preserved portion is only the anterior half of the bone, which appears small and laminar. As in the holotype lower jaw, the anterior mylohyoid foramen is present as a shallow and wide notch on the ventral margin, being completed by the dentary. The smaller and circular anterior inferior foramen pierces the splenial, thus possibly confirming the supposed taphonomical origin of the condition seen in the holotype.

**Dentition.** The dentition of the paratypes (Figs. 3B–3D) shares the same morphology as the holotype. The only possible difference is that, in some specimens, teeth are exposed laterally for more than one third of their height, even though apparently not reaching half of the height exposed. Tooth-number ranges are as follows: 10 to 11 for premaxillary teeth; 26 to 29 for maxillary teeth (24 in the likely juvenile SNSB-BSPG 1970 XVIII 7361); 30 to 37 in dentary teeth (23 in the juvenile SNSB-BSPG 1970 XVIII 7291 and 27 in the juveniles SNSB-BSPG 1970 XVIII 7288 and SNSB-BSPG 1970 XVIII 7290).

## Phylogenetic Analysis

The constrained analysis recovered 66 most-parsimonious trees, with five replications and a length of 784 steps. The strict consensus tree (Fig. 7A; consistency index: 0.552; retention index: 0.33) shows a polytomy at the base of non-eublepharid gekkonoids, including *Palaeogekko*. Resolution can be improved only by pruning *Palaeogekko* itself and *Laonogekko lefevrei* Augé, 2003, even though still maintaining an unresolved Sphaerodactylidae (Fig. 7B). Several alternative positions are possible for the two pruned taxa. A survey of the resulting most-parsimonious trees revealed that in about 18% of the cases (12 trees out of 66) *Palaeogekko* is recovered in a polytomy with Phyllodactylidae + Gekkonidae and Sphaerodactylidae, either alone or in a clade with *Laonogekko* Augé, 2003. Subsequent most-recovered positions are: (1) sister to the clade of non-eublepharid gekkonoids; (2) sister to *Tarentola mauritanica* (Linnaeus, 1758), the only phyllodactylid present in the analysis; (3) sister to *Laonogekko* + Sphaerodactylidae; and (4) crown sphaerodactylid. Each of these cases is recovered in nine trees (about 14%). Less-recovered positions (three trees each: about 5%) are: (1) sister to the clade including Phyllodactylidae + Gekkonidae and Sphaerodactylidae, but in a clade with *Laonogekko*; (2) Sister to *T. mauritanica*, but in a clade with *Laonogekko*; (3) crown gekkonid, in a clade with *Laonogekko* that is sister to *Hemidactylus turcicus* (Linnaeus, 1758); (4) sister to

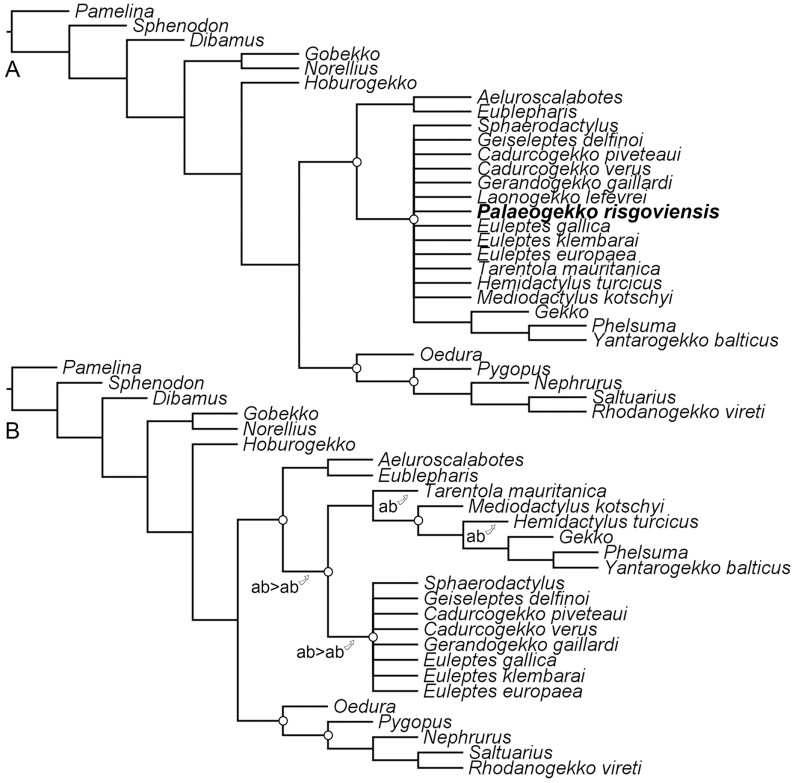

**Figure 7 Results of the phylogenetic analysis with the revised scorings for *Palaeogekko risgoviensis*.** (A) Strict consensus of 66 most-parsimonious trees, with a length of 784 steps. (B) Pruned strict consensus tree, excluding *P. risgoviensis* (a) and *L. lefevrei* (b); arrows mark the possible positions of the pruned taxa. White circles mark the constraints implemented in the analysis (not all constraints are mapped in (A), because some clades are collapsed in a polytomy due to unstable wild-card taxa).

Sphaerodactylidae, in a clade with *Laonogekko*; (5) sister to Sphaerodactylidae, with *Laonogekko* in a more early-branching position; and (6) sister to Sphaerodactylidae, including *Laonogekko*.

## DISCUSSION

### The taxonomic identity of *Palaeogekko risgoviensis*

In their review of fossil gekkotans, *Daza, Bauer & Snively (2014)* raised the matter of multiple species possibly being represented in the type material of *P. risgoviensis*. According to these authors, this was mainly suggested by heterogeneity in the meristic and morphometric data as reported by *Schleich (1987)*. The data presented herein show that the revised measurements and tooth/foramina counts taken on the best-preserved specimens (see Data S1) agree with the ranges observed for single species of extant European geckos by *Villa et al. (2018b)*. Furthermore, general morphology is comparable within all *Palaeogekko* fossils representing the same skeletal element (except of course for the non-gekkotan SNSB-BSPG 1970 XVIII 7262; see Material and Methods section). Variation is indeed present in a few features: (1) development of the dorsal half of the septonasal crest (premaxilla); (2) presence/absence of accessory foramina related to the ones of the

longitudinal canals (premaxilla); (3) development of the anterolateral process (maxilla); (4) shape and length of the posterior notch on the Meckelian fossa (dentary); (5) development of the subdental ridge (dentary); and (6) lateral exposure of the teeth (dentition). For at least some of these features, intraspecific variation is known in extant geckos or other lizards. Different development of the dorsal half of the septonasal crest is reported for lacertid lizards, and a trend of increasing development with growth is known for the anguid *Anguis Linnaeus, 1758* (*Villa & Delfino, 2019a*). Lacertids also display variation in the presence of accessory foramina related to the longitudinal canals in the premaxilla (*Villa & Delfino, 2019a*). The anterolateral process is variable in *M. kotschyi*, which has either a little process or no visible process at all (*Villa et al., 2018b*) in a similar way to *P. risgoviensis*. In most cases, variable features of the German species are mainly represented in one of the different conditions, with the second one being only rarely observed. This is the case for the accessory foramina on the premaxilla, the different shape and length of the posterior notch in the dentary, and the development of the subdental ridge. Missing variation in features like these in extant European gekkotans may be linked with a lower sampling of skeletons of extant animals in osteological works dealing with these reptiles (such as the one by *Villa et al. (2018b)*), compared with the higher number of available fossil specimens for *P. risgoviensis* (in particular as far as the dentaries are concerned). Given the overall similarity in both morphology and size, the six above-mentioned features are herein considered as subject to intraspecific variation, and all the type material of *P. risgoviensis* as pertaining to a single species. A few smaller remains (*e.g.*, SNSB-BSPG 1970 XVIII 7288, SNSB-BSPG 1970 XVIII 7290, SNSB-BSPG 1970 XVIII 7291, and SNSB-BSPG 1970 XVIII 7361) most likely represent juvenile individuals.

**Comparison with extinct European taxa.** *Schleich (1987)* mostly based the description and diagnosis of his new species on meristics and morphometry, but more detailed morphological comparisons can be provided to better understand differences and affinities of *P. risgoviensis* with other extinct and extant European gekkotans (see also Data S3). *Palaeogekko* cannot be compared with *Rhodanogekko vireti Hoffstetter, 1946* and the recently-described *Dollogekko dormaalensis Čerňanský et al., 2022*, however, because these two species are only known from a single isolated frontal each. It cannot be compared with *Yantarogekko balticus Bauer, Böhme & Weitschat, 2005* either, because the latter is preserved in amber. When it comes to other species, for which bones that are also preserved in the material referred to *Palaeogekko* are known, one of the most interesting features to be evaluated is the groove following the last ventrolateral foramen on the maxillae. This is absent in *Laonogekko* (*Augé, 2003*; *Daza, Bauer & Snively, 2014*) and all non-sphaerodactylid extant European gekkotans (*Villa et al., 2018b*) and is one of the main features of euleptine geckos (*Villa, Wings & Rabi, 2022*), being present in all of them but *G. arambourgi* (unknown for *Gerandogekko gaillardi Hoffstetter, 1946*). It is absent in *P. risgoviensis*.

Further differences are present between *Palaeogekko* and euleptines, but there are also shared features. The smooth lateral surface of the maxilla, for example, is shared with *Euleptes* and *Geiseleptes Villa, Wings & Rabi, 2022*, but not *Cadurcogekko* Hoffstetter,

1946. The presence of an anterior point on the facial process is also shared with almost all euleptines (unknown in *Geiseleptes* and *Gerandogekko*), even though the point is longer in *E. europaea* and *Euleptes gallica Müller, 2001* (*Müller, 2001*; *Villa et al., 2018b*) compared to *Palaeogekko*. *Cadurcogekko* clearly differs from *Palaeogekko* also in the presence of a postnarial anterodorsal depression (*Augé, 2005*; *Daza, Bauer & Snively, 2014*; *Bolet et al., 2015*; *Georgalis, Čerňanský & Klembara, 2021*), with *Cadurcogekko piveteaui Hoffstetter, 1946* further presenting, in contrast to *Palaeogekko*, a marked and wide articulation surface with the nasal on the medial surface of the facial process and a lower count of maxillary teeth, dentary teeth, and ventrolateral foramina. The other undisputed Palaeogene euleptine, *Geiseleptes*, is larger than *Palaeogekko* and possesses a longer posterior process of the maxilla and, possibly, a dorsally-shifted posterior surangular foramen (*Villa, Wings & Rabi, 2022*).

Within Miocene euleptines, comparisons between *Palaeogekko* and the two species of *Gerandogekko* is difficult and insufficient, because of the bad preservation of the maxillae and dentaries known for this French genus (*Hoffstetter, 1946*; *Daza, Bauer & Snively, 2014*). At least *G. arambourgi* have a long posterior process of the maxilla with a rounded end (*Hoffstetter, 1946*: fig. 3C; A. Villa, 2022, personal observations), but the single topotypic maxilla is missing part of the anterior half of the bone and further material is needed in order to better evaluate this morphology possibly discriminating the species from *P. risgoviensis*. In this sense, it has to be noted that the only known maxilla of *G. gaillardi* seems to have a shorter posterior process, even though its distal end is broken off (*Hoffstetter, 1946*: fig. 4B; *Daza, Bauer & Snively, 2014*: fig. 7C, 7D). In both the maxillae of *G. arambourgi* and *G. gaillardi*, the facial process is not preserved, further complicating the comparison with *P. risgoviensis*. It is easier, on the other hand, to differentiate *Palaeogekko* from all species of *Euleptes*. The most extensive comparison possible is of course with the extant *E. europaea*, which displays a set of characters that allow to discriminate each skeletal element known for *P. risgoviensis* from the corresponding ones of the European leaf-toed gecko. Apart for the already mentioned presence of the groove following the last ventrolateral foramen in the extant species, these include (*Villa et al., 2018b*): (1) the arrow-shaped ascending nasal process of the premaxilla; (2) the long and posteriorly-rounded posterior process of the maxilla; (3) the absence of a distinct carina maxillaris; (4) the longer extension of the posterior notch of the Meckelian fossa in the dentary; (5) the complete closure of the anterior mylohyoid foramen; (6) the ventral bending of the posterior end of the splenial; (7) the rounded end of the coronoid process; (8) the rounded distal end of the posteromedial process of the coronoid; and (9) the dorsally-shifted posterior surangular foramen. Features 1 and 4 in this list apply to *E. gallica* as well, when this extinct species is compared with *Palaeogekko*. In addition, tooth count is slightly higher in the premaxilla and slightly lower in the maxilla of *Palaeogekko* as compared to *E. gallica* (*Müller, 2001*; *Čerňanský & Bauer, 2010*; *Daza, Bauer & Snively, 2014*). In contrast to *E. europaea*, *Palaeogekko* shares with both extinct *Euleptes* species, *E. gallica* and *Euleptes klembarai Čerňanský, Daza & Bauer, 2018*, the presence of a carina maxillaris (*Müller, 2001*; *Čerňanský & Bauer, 2010*; *Daza, Bauer & Snively, 2014*; *Čerňanský, Daza & Bauer, 2018*), which is however more distinct, straight

and more posterodorsally directed in the two extinct *Euleptes*. A peculiar combination of characters is apparently displayed by the material referred to *Euleptes* sp. from the Oppenheim/Niersten quarry, in Germany, which include arrow-shaped premaxillae with eight or 10 teeth and at least one maxilla with a sigmoid carina maxillaris more similar to the one of *Palaeogekko* than to other extinct *Euleptes* (at least based on the drawings provided by *Müller & Mödden (2001)*: fig. 1). The rest of the maxilla seems rather comparable with *Palaeogekko* as well (*i.e.*, short and pointed posterior process, smooth lateral surface), but the anterior point of the facial process appears longer and a posterior groove follows the last ventrolateral foramen.

The last fossil species that can be compared with *P. risgoviensis* is *L. lefevrei*. Similar to *Palaeogekko*, the maxilla of *Laonogekko* has no groove associated with the last ventrolateral foramen, a short anterior point of the facial process, a smooth lateral surface, and a sigmoid carina maxillaris (*Augé, 2003*, *2005*; *Daza, Bauer & Snively, 2014*). However, *Palaeogekko* clearly differs from this Eocene species based on: (1) a steeper posterior margin of the facial process; (2) the absence of a marked articulation surface with the nasal on the medial surface of the facial process ("inner ledge" *sensu Daza, Bauer & Snively, 2014*); and (3) a lower number of maxillary teeth. In the original descriptions made by *Augé (2003*, *2005)*, *Laonogekko* also had a higher number of dentary teeth, but *Daza, Bauer & Snively (2014)* estimated a lower tooth count for the most preserved dentary, which fits the range of *P. risgoviensis*.

**Comparison with extant European taxa.** In a similar way to *E. europaea*, other extant, non-sphaerodactylid European geckos also allow more detailed comparisons with *Palaeogekko* than the mentioned extinct species. Both differences and similarities can be highlighted, based on the morphological data reported by *Villa et al. (2018b)*. *Palaeogekko* differs from all *H. turcicus*, *M. kotschyi*, and *T. mauritanica* in having: (1) a shallow notch separating the palatal processes of the premaxilla (this feature is shared with *E. europaea*); and (2) a shallow expansion at midheight of the ascending nasal process. The ascending nasal process is also not expanded at the distal end as it is in *M. kotschyi*. The maxilla of the German species has a well-developed and pointed anteromedial process like *T. mauritanica* (and *E. europaea*), but not *H. turcicus* and *M. kotschyi*. The anterior margin of the facial process is inclined anteriorly in *P. risgoviensis*, *E. europaea*, and *H. turcicus*, and vertical in *M. kotschyi*. In *T. mauritanica*, it presents a small notch, unlike *P. risgoviensis*. The smooth lateral surface is shared with all species (including *E. europaea*), but *H. turcicus*. The maxilla of the latter species also differs from the one of *Palaeogekko* in the presence of the lacrimal groove and the absence of a strong anterior point on the facial process, but they similarly display a distinct carina maxillaris. Nevertheless, the latter is more vertically oriented in *P. risgoviensis* than in *H. turcicus*, whose carina is posterodorsally directed. As far as the dentary is concerned, the extension of the posterior notch of the Meckelian fossa is longer in *H. turcicus*, but similar to *Palaeogekko* in *M. kotschyi* and *T. mauritanica*. The dentary of *T. mauritanica* differs from that of *P. risgoviensis*, however, because the anterior opening of the Meckelian fossa is not continued into a groove in the phyllodactylid. The splenial of *P. risgoviensis* is comparable with the one of *M. kotschyi* in both the anterior mylohyoid foramen present as a notch on the ventral margin of the bone and the posterior

**Table 1 Tooth density indexes (number of tooth position per mm) for *P. risgoviensis* and European extant gekkotans.**

|  | Premaxilla | | Maxilla | | Dentary | |
|---|---|---|---|---|---|---|
|  | Min | Max | Min | Max | Min | Max |
| *P. risgoviensis* | 4.5 | 5.3 | 4.2 | 5.3 | 4.3 | 5.9 |
| *E. europaea* | 5.3 | 6.4 | 5.3 | 6.6 | 5.8 | 7.4 |
| *H. turcicus* | 5.3 | 5.5 | 4.4 | 5 | 4.1 | 4.8 |
| *M. kotschyi* | 4.7 | 5.8 | 4.2 | 4.9 | 4.9 | 5.5 |
| *T. mauritanica* | 3 | 4.1 | 2.8 | 3.8 | 3 | 3.4 |

**Note:**
Values for the extinct German species are based on the most complete specimens. This, together with differences in the revised measurements and tooth counts, explains the differences between ranges presented here and those reported by *Schleich (1987)*. Values for the single fossils are available in the Data S1. Data for extant species come from personal observations on specimens listed by *Villa et al. (2018b)*.

end of the bone not bending ventrally. Both these features are different in *H. turcicus*, whereas *T. mauritanica* has a splenial with a bending posterior end but can present both a notch-like or, more rarely, completely-closed anterior mylohyoid foramen. In the end, *P. risgoviensis* shares with both *H. turcicus* and *M. kotschyi* a pointed posteromedial process of the coronoid and a posterior surangular foramen not shifted dorsally. Both these features are different in *T. mauritanica*, which shows a rounded process and a shifted foramen.

In terms of measurements and meristic characters, *Palaeogekko* compares with *H. turcicus* and *M. kotschyi* in premaxillary width and tooth-row length of both maxillae and dentaries, whereas it is larger than *E. europaea* and smaller than *T. mauritanica*. It has more teeth than *E. europaea* in the premaxilla, but a similar number to other European extant geckos. Total counts for maxillary and dentary teeth are comparable with all four extant species. Tooth density indexes (Table 1) for premaxillae, maxillae, and dentaries are lower in *Palaeogekko* than in *E. europaea*, higher than in *T. mauritanica*, and comparable with *H. turcicus* and *M. kotschyi*, thus following an inverted pattern then the one shown by measurements.

***Palaeogekko risgoviensis* as a distinct species.** Putting together all the information presented above, *Palaeogekko* clearly stands out as different from almost all European geckos with which it can be compared, both extinct and extant. Uncertainties remains only on the characters possibly discriminating this Middle Miocene German species with the French *Gerandogekko*, remains of which come from Lower and Upper Miocene deposits (*Hoffstetter, 1946*; *Daza, Bauer & Snively, 2014*). This is mainly due to the facts that the most significant fossils of both *Gerandogekko* species are frontals, an element that is unknown for *Palaeogekko*, and that maxillae and dentaries of the French species are insufficiently preserved. The different morphologies shown by the posterior processes of the maxillae of *G. arambourgi* and *G. gaillardi* further complicates this situation, given that only the long and rounded process of the former undisputedly differs from the condition observed in maxillae of *Palaeogekko*. Whether this means that *P. risgoviensis* is somehow related with at least *G. gaillardi* or even with *Gerandogekko* as a whole cannot be stated pending the recovery of further fossils (*i.e.*, a frontal referrable to *P. risgoviensis* or

better-preserved maxillae of the two *Gerandogekko* species). For the time being, *P. risgoviensis* can be maintained as a distinct species, living in central Europe in a moment (MN 6) when only one other gecko is currently known to have been present: the Slovakian euleptine *E. klembarai*.

## Phylogenetic relationships of *Palaeogekko risgoviensis*

Available data and phylogenetic data matrixes still fail to confidently clarify the phylogenetic relationships of *P. risgoviensis*. The German species is recovered in several possible positions within the gekkotan tree by the phylogenetic analysis herein presented, including as a stem non-eublepharid gekkonoid, a possible phyllodactylid (or at least related to the only phyllodactylid in the analysis, *T. mauritanica*), a crown gekkonid, and either a stem or crown sphaerodactylid. Thus, the new analysis concurs with *Daza, Bauer & Snively (2014)* in the non-eublepharid gekkonoid nature of *Palaeogekko*, without being able to further disentangle its relationships within this clade in a confident way.

The potential stem-pygopodoid topology recovered by *Villa, Wings & Rabi (2022)* is not emerging here, most likely as a result of the revised scorings. These new results agree with the current south-Pacific distribution of pygopodoids and the absence of any other convincing evidence of either stem- or crown-pygopodoid presence outside from their modern range (pygopodoid affinities recovered for *R. vireti* here and by *Villa, Wings & Rabi (2022)*, are most likely due to convergence and overall poor knowledge of this taxon; see also *Daza, Bauer & Snively, 2014*). Similar to *Villa, Wings & Rabi's (2022)* analysis, on the other hand, a certain link between *Palaeogekko* and *Laonogekko* is revealed. Considering the striking time span separating the two taxa (Middle Miocene *vs* early Eocene, respectively) and the clear morphological differences, this possible relation is worth of further investigations in future works.

Partly in contrast to the results of the phylogenetic analysis, comparisons with other European fossil geckos seem to exclude affinities of *Palaeogekko* with euleptine sphaerodactylids. Significant differences can be highlighted, and most similarities are represented by features shared with taxa belonging to other clades (*e.g.*, the smooth lateral surface shared with *Euleptes* and *Geiseleptes*, but also *Mediodactylus* and *Tarentola*; the presence of a carina maxillaris shared with extinct *Euleptes* and *Hemidactylus*). The same can be told for the comparison with the extant *E. europaea*. A significant exception is *Gerandogekko*, for which available information are insufficient for a detailed comparison. This is unfortunate, because *Gerandogekko* is currently the only undisputed euleptine devoid of a groove following the last ventrolateral foramen of the maxilla, the same condition seen in *Palaeogekko*. *Gerandogekko* could, thus, represent an important taxon to verify possible euleptine affinities of *Palaeogekko*, and whether members of the clade with ungrooved maxillae where more widespread during the Miocene.

Evaluating affinities with other clades is hampered by the lack of extinct species that can be directly compared with *Palaeogekko*. Some information can be retrieved from comparisons with extant species, but these face a limit in the fact that at least some modern European populations of non-sphaerodactylid geckos are interpreted as recent colonizers (*H. turcicus*: *Carranza & Arnold, 2006*; *Rato, Carranza & Harris, 2011*; *T. mauritanica*:

*Harris et al., 2004*; *Rato et al., 2010*) and may thus not be related with a Miocene species. Fossil evidence of a previous occupation of Europe by at least some of these is also available, however (*Villa & Delfino, 2019a*). *Schleich (1987)* already pointed out affinities of his new species with *Tarentola* and *Mediodactylus*, even though without discussing them in detail. *Palaeogekko risgoviensis* indeed presents similarities with these taxa, as well as with European *Hemidactylus*. Significant differences are also present, though (see comparisons above). Simply based on the number of shared characters and differences observed, the most comparable among non-sphaerodactylid extant European species seems to be *M. kotschyi*, but again, there is no character uniquely shared with this species and differences are also evident. It is, therefore, clear that the phylogeny of *P. risgoviensis* is still far from being resolved, and new data and analyses are greatly anticipated.

## CONCLUSIONS

The redescription and reevaluation of the type material of *P. risgoviensis* supports the status of the Middle Miocene German gecko as a valid species, which can be differentiated from almost all other known extant and extinct geckos in Europe. With the single exception of a dentary pertaining to another lizard group, there is also no evidence of multiple species being represented in the type material, suggesting a particular abundance of this gecko at the Steinberg locality. Available data are not sufficient to make confident inferences on the phylogenetic relationships of this taxon, however, in particularly due to the absence of preserved frontals. The question still remains open, therefore, on whether or not *P. risgoviensis* could be evidence of the presence of non-euleptine geckos in Europe already during the Miocene. Possible relationships with the Eocene *Laonogekko* suggested by the phylogenetic analysis are interesting and worth of further scrutiny. European geckos were already demonstrated as one of the few clades including inhabitants of the continent that persisted with the same lineage from the Palaeogene to the Neogene (with euleptines; *Villa, Wings & Rabi, 2022*), and a link between *Laonogekko* and *Palaeogekko* may further add on this pattern. However, this is still covered with uncertainty, given the combination of strongly unstable phylogenetic results and amount of missing data regarding the two taxa. Morphological affinities with *Mediodactylus*, on the other hand, also deserve further investigation, because this is the only extant non-euleptine gekkotan genus whose European population (currently referred to several different species) started to differentiate and possibly to colonize the continent already during the Miocene (*Kotsakiozi et al., 2018*). Additional work on the phylogeny and evolutionary history of European gekkotans, combining data on both extant animals and fossils, are greatly anticipated. The present contribution will allow to better include the enigmatic *P. risgoviensis* in these new studies as well.

## ACKNOWLEDGEMENTS

The present study benefited from useful discussions with David Alba, Arnau Bolet, Alessandro Urciuoli, and Evangelos Vlachos. Oliver Rauhut and Massimo Delfino are thanked for access to fossils stored in the SNSB-BSPG and comparative specimens in the Università degli Studi di Torino, respectively. Victor Beccari kindly provided the photos of

SNSB-BSPG 1970 XVIII 7262 used for Fig. 1. The Willi Hennig Society is thanked for making the TNT software freely available online. I would also like to acknowledge the Academic Editor of PeerJ, Michela Johnson, and the three reviewers, Márton Rabi, Davit Vasilyan, and Georgios Georgalis, for helping improving this article with their useful comments. This is publication number 365 of the Museum of Geology and Palaeontology of the Università degli Studi di Torino.

### Funding

The author was supported by a Humboldt Research Fellowship from the Alexander von Humboldt Foundation during part of the development of this work. He is now funded by the Agencia Estatal de Investigación, with a Juan de la Cierva-Formación grant (FCI2019-039443-I/AEI/10.13039/501100011033). The funders had no role in study design, data collection and analysis, decision to publish, or preparation of the manuscript.

### Grant Disclosures

The following grant information was disclosed by the authors:
Humboldt Research Fellowship from the Alexander von Humboldt Foundation.
Agencia Estatal de Investigación, with a Juan de la Cierva-Formación: FCI2019-039443-I/AEI/10.13039/501100011033.

### Competing Interests

The author declares that he has no competing interests.

### Author Contributions

- Andrea Villa conceived and designed the experiments, performed the experiments, analyzed the data, prepared figures and/or tables, authored or reviewed drafts of the article, and approved the final draft.

### Data Availability

  Raw measurements are available in the Supplemental Files.

### Supplemental Information

Supplemental information for this article can be found online at http://dx.doi.org/10.7717/peerj.14717#supplemental-information.

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
