# Peer review of "A redescription of Palaeogekko risgoviensis (Squamata, Gekkota) from the Middle Miocene of Germany, with new data on its morphology"

_PeerJ, doi:10.7717/peerj.14717_

## Round 0.1 · original submission · Minor Revisions

The manuscript reads well and the figures are well done. The comments for the manuscript were overall positive - the reviewers mainly suggest compacting/shortening certain areas of the manuscript, in addition to checking the reference list and small grammatical errors. Also, make clear if you are following the PhyloCode (comment by M. Rabi) - if so, the rank(s) will need to be registered, if not already, in RegNum.

·

Basic reporting

In the introduction, please briefly elaborate on why a re-description is necessary (e.g., the previous description was not detailed enough, our knowledge of comparable taxa significantly improved since the time of publication, specimens need to be better illustrated, different interpretation of morphology etc.). I suggest making the rest of the introduction more compact and focus on how this study is filling a gap in our knowledge. I realize that presently all the information in the introduction is more or less relevant but summarizing these in a more compact way would better clarify the scope of your study.

In the abstract, please specify that your study includes a phylogenetic analysis (rephrase).

Under Materials and Methods, please indicate where the phylogenetic matrix is deposited.

Discussion: The comparison part is also on the lengthy side and could be more compact. At least try to break it up using subheading (e.g. by taxa), otherwise difficult to follow and is too crowded.

Experimental design

"Same as in the as in the second iteration of the..." Delete "same".

For the sake of clarity, please specify that the constraints used in the phylogenetic analysis are of a molecular topology and add reference.

Validity of the findings

No comment.

Additional comments

I find this a thorough, high quality, and absolutely relevant work.

Other minor comments:

In contrast to instead of with

Far from being resolved instead of been resolved

Villa et al. (2022 ab) not consistently cited.

Systematic paleontology: by not using ranks you make the impression that you follow the PhyloCode. If so all defined clade names should be in Italics and please explicitly comment on the nomenclature you are adopting. If you don't follow the PhyloCode then you should use ranks. But then why using the clade name Euleptinae (a phylogenetically defined clade)?

·

Basic reporting

I suggest to include in the line 119-120 "Family incertae sedis"
ll. 296-298. is there a similar intraspecific variability documented in the recent or any other fossil species. I find it imporant to discuss a bit more this variability.
l. 363. "fossils" should be in singular
l. 484 replace "Palaeogene" by "Paleogene".

Experimental design

all correct.

Validity of the findings

fully

·

Basic reporting

All my comments are presented in section "4. Additional comments".

Experimental design

All my comments are presented in section "4. Additional comments".

Validity of the findings

All my comments are presented in section "4. Additional comments".

Additional comments

The manuscript of Villa present a comprehensive redescription of Palaeogekko risgoviensis – although this taxon represents one of a few named fossil gekkotans from Europe, it has curiously remained so far inadequately known. The description is much informative and detailed; this, coupled with the large and nicely prepared figures, allows a more in depth and conclusive knowledge on this taxon. I have only some minor remarks on the text, mostly regarding typification issues and some missing relevant references:
Line 25: Here I would suggest citing Georgalis and Scheyer (2021) who provided a discussion on the earliest Miocene herpetofaunas of Europe:
Georgalis, G.L. and T.M. Scheyer. 2021. Lizards and snakes from the earliest Miocene of Saint-Gérand-le-Puy, France: an anatomical and histological approach of some of the oldest Neogene squamates from Europe. BMC Ecology and Evolution 21:144.
Line 36: the vernacular term is spelled “chameleons” (without “a”).
Line 52: You mean Sphaerodactylidae as a whole, I suppose (and not Euleptes). Because the Eocene form, Geiseleptes, is a sphaerodactylid indeed but we cannot be certain that it is a direct descendant of the Neogene Euleptes or whether spaherodactylids became extinct in the Paleogene and reappeared later with other representatives (e.g., like in the case of Varanidae, where in the Eocene there was Saniwa and the Neogene forms (i.e., Varanus) are the product of another dispersal event). So maybe some rephrase here.
Lines 88-89: You cannot do this. Once designated as a paratype, it always remains a paratype. Unless you make a formal petition to ICZN. But I see no problem here anyway: paratypes have no nomenclatural power and do not affect the validity of a species (only holotypes, lectotypes, syntypes) can do so. So you can simply leave it as a paratype and clearly state that this paratype does not pertain to this species (this does not make the taxon a chimaera – it would be so, if this specimen was a syntype). Besides, has this specimen “7262” been previously figured before? If not, it would be good if you present it here as a supplementary figure (perhaps) and in any case to provide an emended taxonomic identification for this, e.g. Squamata indet., or even if you can go even more precisely to some family level referal.
Line 123: Insert also in the list this other paratype, which you identified as non-Palaeogekko (and mention in a parenthesis that it does not pertain to the species).
Line 301: Schleich (1987)
Line 316: For Cadurcogekko, I suggest citing Georgalis et al. (2021) who presented a new maxilla from Quercy:
Georgalis, G.L., A. Čerňanský and J. Klembara. 2021. Osteological atlas of new lizards from the Phosphorites du Quercy (France), based on historical, forgotten, fossil material. Geodiversitas 43(9):219–293.
Line 321: Again, for Cadurcogekko it is better if you mention here Georgalis et al. (2021), who, besides the new specimens, also clarified taxonomic and nomenclatural issues of the genus.
Line 423: “only one other...”
I am sure these few suggestions can be easily be implemented in the manuscript and therefore I am recommending it for publication after such minor revision.
I am at your disposal for any further query.
Yours sincerely
Dr. Georgios Georgalis

---

## Round 0.2 · accepted · Accept

Congratulations on the manuscript. All editor and reviewer comments/suggestions have been met and noted. The revision has been assessed by the academic editor, who finds the current version acceptable. The author just needs to (1) double check spelling and grammar (extremely minor) in the manuscript, and (2) note in Materials & Methods that the phylogenetic analysis conducted was an unweighted maximum parsimony and how many replications were done in the New Technology Search (for clarification). The author can also choose to include Bremer support values in the analysis (not necessarily needed).